# DECOMPOSITION INTO INVARIANT SPACES WITH $L_1$-TYPE CONTRASTIVE LEARNING

## ABSTRACT

Recent years have witnessed the effectiveness of contrastive learning in obtaining the representation of dataset that is useful in interpretation and downstream tasks. However, the mechanism by which the contrastive learning succeeds in this feat has not been fully uncovered. In this paper, we show that contrastive learning can uncover a fine decomposition of the dataset into a set of latent features defined by augmentations, and that such a decomposition can be achieved just by changing the metric in the simCLR-type loss.

## 1 INTRODUCTION

Collectively, contrastive learning refers to a family of representation-learning methods with a mechanism to construct a latent representation in which the members of any *positive pair* with similar semantic information are close to each other while the members of *negative pairs* with different semantic information are far apart (Hjelm et al., 2019; Bachman et al., 2019; Hénaff et al., 2020; Tian et al., 2019; Chen et al., 2020). Recently, numerous variations of contrastive learning (Radford et al., 2017; Li et al., 2021; Wang et al., 2021b; Joseph et al., 2021; Laskin et al.) have appeared in literature, providing evidences in support of the contrastive approach in real world applications. However, there still seems to be much room left for the investigation of the reason why the contrastive learning is effective in the domains like image-processing.

Recent works in this direction of research include those pertaining to information theoretic interpretation (Wang & Isola, 2020) as well as the mechanism by which the contrastive objective uncovers the data generating process and the underlying structure of the dataset in a systematic way (Zimmermann et al., 2021). In particular, von Kügelgen et al. (2021) has shown that, under a moderate transitivity assumptions with respect to the actions of the augmentations used in training, the contrastive learning can isolate the *content* from the *style*—where the former is defined as the space that is fixed by all augmentations and the latter is defined as the space altered by some augmentation. Meanwhile, Wang et al. (2021a) described the similar philosophy in terms of group theoretic context, proclaiming that the contrastive learning can decompose the space into inter-orbital direction and intra-orbital direction. They even went further to decompose the inter-orbital direction by introducing an auxiliary IRM-type loss. In a related note, Fumero et al. (2021) assume that the dataset of interest has the structure of a product manifold with the assumption that each augmentation family alters only one of the products. They train a set of nonlinear projection operators to extract each component of the manifold in a self-supervised way.

In this research, we present a result suggesting that contrastive learning alone can possibly decompose the data space not just into the common-invariant space of all augmentations and its complement, but also into finer spaces that can generate the invariant space of *each* augmentation. We empirically show that a standard contrastive learning can reveal such a decomposition of the space by simply changing the metric from the commonly used angle distance $d(a,b) = a^T b$ to the Lasso-type metric like $d(a,b) = |a-b|$. Moreover, we show that, in a special case, this decomposition of the space can be block-identified without assuming strict group structure on the set of augmentations.

## 2 METHOD

Let $\mathcal{X}$ be the space of dataset, and let $\mathcal{T} = \{T_i : \mathcal{X} \to \mathcal{X}\}$ be a set of augmentations. Suppose

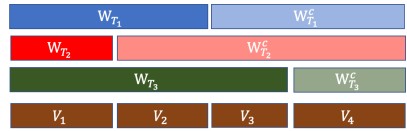

Figure 1: First three rows: the decomposition of the latent space of $\mathcal{X}$ induced by three different augmentations, $T_1, T_2$ and $T_3$. The space $W_T$ is the invariant space of $T$. The bottom row: the decomposition of $\mathcal{X}$ into frequency components $\{V_k\}$ that can generate any intersections of $W_T$s.

that $h$ is an invertible map from $\mathcal{X}$ to some latent space. We use $W_T \subset h(\mathcal{X})$ to denote the latent invariant space of a specific augmentation $T$, i.e., $h \circ T \circ h^{-1}(w) = w$ for any $w \in W_T$. We would like to consider a decomposition of the space defined with $W_T$s. The first three rows in Figure 1 are the visualization of the decomposition of $\mathcal{X}$ into invariant/non-invariant space for three different choices of augmentations, $T_1, T_2$ and $T_3$. When there are three invariant spaces as such, we want to consider the decomposition of the space into $\{V_k\}$ defined as the minimal intersections of invariant spaces and their complements (fourth row).

Our decomposition has a clear difference from the relevant works; von Kügelgen et al. (2021) showed that contrastive learning can decompose $\mathcal{X}$ into the space $V_{content}$ that is invariant under the action of all members of $\mathcal{T}$ and its complement $V_{style}$. Because $V_{content}$ is our definition of $V_1$, the intersection of $W_{T_1}, W_{T_2}$ and $W_{T_3}$, the structure discussed in von Kügelgen et al. (2021) is only a decomposition of the space into $V_1$ and $\bigoplus_{j \neq 1} V_j$. Meanwhile, Wang et al. (2021a) discusses a further decomposition of $V_1$, but not the other parts.

The most important property of $\{V_k\}$ is that it can generate the intersection of the invariant spaces of any arbitrary subset of $\{T_i\}$. Thus, if $h$ is such that each $V_k$ can be represented by a set of its coordinates, then $h$ can represent any intersection of invariant spaces as a set of coordinates. As the concept we introduce here is akin to the one introduced in harmonic analysis of groups (Weintraub, 2003; Garsia & Ömer Egecioglu, 2020; Clausen & Baum, 1993), we would like to say that the latent space is *Symmetry Adapted* to $\mathcal{T}$ if the intersection of the invariant spaces for any pair $(T, T') \subset \mathcal{T}$ can be represented by a set of coordinates[1]. Likewise, we would like to call the smallest intersection components (e.g. $V_k$s) that make up the invariant space of each $T$ as a *frequency*. It turns out that, at least in a case in which every frequency can be expressed as an intersection of invariant spaces, such a decomposition can be identified uniquely up to a mixing within each frequency. The result below is an informal statement of this claim, and is an analogue of the block-identifiability results in von Kügelgen et al. (2021).

**Proposition 1** (Informal). *Suppose that $\{V_j\}$ consists of minimal intersections of invariant spaces $\{W_T\}$, and suppose that there exists an invertible map $h : \mathcal{X} \to \oplus V_j$. If $\mathcal{T}$ satisfies a certain transitivity assumption then $\{V_j\}$ are block identifiable. That is, if there is any other such $\tilde{h}$ with $\{\tilde{V}_j\}$, then there exists an invertible map between $V_j$ and $\tilde{V}_j$.*

We note that this statement assumes that the set of augmentations defining the frequencies is rich enough that there is no frequency that is not fixed by any augmentation. In a related note, our identifiability result does not necessarily generalize the result of von Kügelgen et al. (2021) because the style space in their work may contain a feature altered by all elements of $\mathcal{T}$. Also, Fumero et al. (2021) assumes that each augmentation alters an unique decompositional component that does not intersect with the space altered by other augmentations. For the formal statement of Proposition 1, see Appendix A.

In this paper, we propose a simple contrastive learning objective with the goal of finding a Symmetry Adapted latent space for $\mathcal{X}$. As shown in Wang & Isola (2020), the contrastive learning based on noise contrastive error (e.g simCLR) can be described as a combination of two losses: (1) the alignment loss that attracts the positive pairs in the latent space and (2) the uniformity loss that encourages the latent variables to be distributed uniformly, thereby preventing the degeneration. We use the loss of the same type, except that we replace the cosine distance norm in simCLR with the

---

[1]In representation theory, the basis of a representation space is called Symmetry Adapted when any irreducible representation of the space can be represented by a subset of basis.

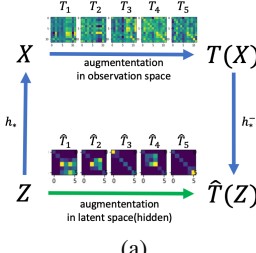 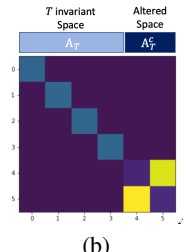

(a)                  (b)

Figure 2: (a) Relation between the augmentation $\hat{T} = h_* T h_*^{-1}$ in the latent space and augmentation $T$ in the observation space. In a symmetry adapted latent space, each augmentations is a direct sum of identity map and a block. (b) For each $\hat{T}$, the identity map part corresponds to invariant space $W_T$, and the block part corresponds to $W_T^c$.

$L_1$ loss. That is, we consider the following objective to train the encoder $h_\theta$ parametrized by $\theta$:

$$L_{align} + L_{anti-deg} = \underbrace{\mathbb{E}_{T_1, T_2, X}[\|h_\theta(T_1(X)) - h_\theta(T_2(X))\|_1 / \tau]}_{L_1 \text{ alignment Loss}} + \underbrace{(-H(h_\theta))}_{\text{Anti-Degeneration Loss}}, \quad (1)$$

where $X$ is the observation input, $T_1$ and $T_2$ are the random augmentations, $\tau$ is the temperature, and $H$ is a function that prevents degeneration, such as Shannon's entropy. Essentially, this objective function differs from simCLR only in the choice of the metric used to bring the positive pairs together; the equation 1 becomes simCLR when we replace $\|a - b\|_1$ with $a^T b$. Notice that $L_1$ distance is particularly different from the angular distance in that is not invariant to the orthogonal transformation, and hence is able to align particular set of dimensions. The gist of this loss is to maximize the number of dimensions at which the pair $(T_1(X), T_2(X))$ agrees in the latent space in the way of LASSO (Tibshirani, 1996). If $\mathcal{T}_1 \subset \mathcal{T}$ has a common invariant space $W_{\mathcal{T}}$ and $\mathcal{T}_2 \subset \mathcal{T}$ has $W_{\mathcal{T}_2}$, and if $\mathcal{T}_1$ and $\mathcal{T}_2$ contains the identity, we can expect our loss to seek a representation in which $W_{\mathcal{T}_1} \cap W_{\mathcal{T}_2}$ as well as $W_{\mathcal{T}_2}$ and $W_{\mathcal{T}_2}$ are maximized. In the next subsection, we study if we can experimentally recover such a decomposition space into $V_i$s using our $L_1$ contrastive loss.

## 3 EXPERIMENTS

### 3.1 LINEAR CASE

Our experiment of a linear case is instrumental in describing the effect of the $L_1$-contrastive loss in the equation 1. In this experiment, we assume that the dataspace $\mathcal{X}$ is a 6 dimensional linear subspace embedded in a 20 dimensional input space via a linear encoding function $h_* : \mathcal{X} = \mathbb{R}^{12} \to \mathbb{R}^6 = \mathcal{Z}$. This setup is a linear analogue of a 6 dimensional manifold embedded in a 12 dimensional ambient space. Suppose that, in the 6 dimensional latent space, each augmentation $T$ acts on the input $z \in \mathcal{Z}$ by mixing a subset of coordinates $A_T \subset \{1, ..., 6\}$ while leaving its complementary coordinates $A_T^c$ fixed. That is, in the notation of the previous section, $R^{A_T} = W_T$. We assume the augmentations in the latent space are also linear, and that at least one symmetry adapted basis exists with the encoder $h_*$. This situation is realized when each augmentation in the latent space, $\hat{T} = h_* T h_*^{-1}$, is block diagonal. See Figure 2 for the visualization of this setup.

Notice that when each augmentation $T$ has an invariant space $W_T = \mathbb{R}^{A_T}$ of its own, the only choices of the coordinate system that can express the invariant space of every $T$ as a "subset of coordinates" is the family of coordinates that can not only express $A_T$ as a subset but also $A_T \cap A_{T'}$ as a subset for all $(T, T')$. However, hidden in the shades of unknown encoder map $h_*$, such an underlying structure may be hard to obtain just with the pairs $(T(X), T'(X))$.

In our linear experiment, we trained an encoder $h_\theta$ with our $L_1$ contrastive loss on the set of $(T(X), T'(X))$ pairs. We chose a diverse enough set of $T$ in this experiment so that, in the hidden symmetry adapted space, each $V_k$ is one dimensional. Thus, the goal of this experiment is to learn an encoder $\hat{h}_\theta^{-1}$ that identifies $h_*$ up to a permutation of the coordinates and scaling. We trained $\hat{h}_\theta$

by our $L_1$ contrastive loss with the reconstruction-error as the anti-degeneration loss; more specifically, we simultaneously trained the decoder function $\hat{h}_\theta^{-1}$ together with $\hat{h}_\theta$. We emphasize that, during our training, neither the choices nor the functional forms of the used $T$s are known to the trainer. Figure 3 illustrates the result of our training. The matrices in the first row of are $h_* T h_*^{-1}$

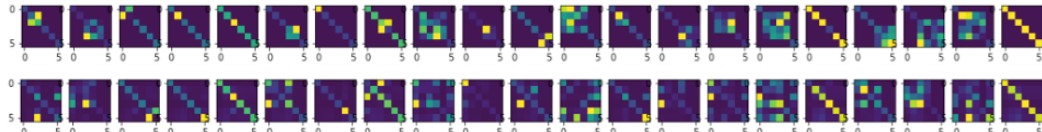

Figure 3: First row: $\hat{T} = h_*^{-1} T h_*$ for the set of $T$s used in the training of $h_\theta$. In the first row, all diagonal entries not belonging to blocks are 1. Second row: $\hat{h}_\theta T \hat{h}_\theta^{-1}$ for the same set of $T$s with $\hat{h}_\theta$ estimated from the $L_1$ contrastive loss. All diagonals not belonging to blocks are approximately 1 (0.946±0.042). Because the size of the blocks and hence the size of invariant spaces are same for the first and the second row, our trained $\hat{h}_\theta^{-1}$ identifies the symmetry adapted space up to permutation.

with $T$s used in our training, and the matrices in the second row are $\hat{h}_\theta T \hat{h}_\theta^{-1}$ obtained with trained $\hat{h}_\theta$. We see that the size of the blocks in the second row are almost the same as the blocks in the blocks in the first row. Also, all diagonal entries not belonging to the estimated non-trivial blocks are approximately 1 (0.946 ± 0.042). Thus, $\hat{h}_\theta$ identifies all six $V_i$s up to a permutation. In other words, we have successfully learned the Symmetry-Adapted basis that can represent the invariant spaces of all augmentations as *sets* of coordinates.

## 3.2 NONLINEAR CASE

To verify if our method can handle a case in which $h_*$ is nonlinear, we experimented on stylized MNIST, a modified version of MNIST in which the digits are randomly colored and rotated.

Each image in this dataset is $32 \times 32 \times 3$ dimensional. We conducted the self-supervised learning

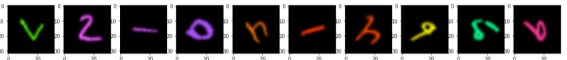

Figure 4: Stylized MNIST dataset

with our $L_1$ contrastive loss on this dataset in hope to learn the latent space that is symmetry adapted to the color-jittering and rotation. In order to facilitate the learning of the decomposition, we made the latent space to be of tensor-form $\mathbb{R}^{d_a \times d_s}$, normalized each $d_s$ dimension, and used group lasso distance for each $z^{(1)}, z^{(2)} \in \mathbb{R}^{d_a \times d_s}$; that is, we used $d(z^{(1)}, z^{(2)}) = \sum_{k \in d_a} \|z_k^{(1)} - z_k^{(2)}\|_2$, where $z_k^{(i)}$ is the $k$th row of $z^{(i)}$ (Yuan & Lin). For this experiment, we selected $dim_s = 32$ and $dim_a = 16$, yielding the total of 512 dimensional latent space. Just like in the convention of normalization in contrastive learning, we normalized each one of 16 rows in the vector. We trained the encoder with ResNet and set the temperature $\tau$ of the contrastive learning to be 0.0001. Please see Appendix C for the detail of the encoder architecture . Our intuition dictates that these two families of augmentations are orthogonal in the sense that the latent subspace altered by the member of the former family is disjoint from the latent subspace altered by the latter family. And indeed, this turns out to be exactly what we observe in the space learned by our $L_1$ contrastive learning. The figure 5 illustrates the vertically stacked 512 dimensional horizontal vectors of $|h_\theta(T(X)) - h_\theta(X)|$ for random $T$. The figure in the first row is produced from randomly rotation transformation $T_{rot}$, and the figure in the second row is produced from randomly color transformation $T_{color}$. As we can see in the the figures, the dimensions that are altered by $T_{rot}$ are not altered by $T_{color}$, and vice versa. Moreover, we can observe the dimensions that are fixed by both $T_{rot}$ and $T_{color}$ as well. We can also see that almost all 512 dimensions are used in the representation. To measure the extent to which the space altered by rotations is complementary to the space altered by the color transformations, we measured the following value

$$\overline{\mathbb{E}_{T_{rot}, X}}[|h_\theta(T_{rot}(X)) - h_\theta(X)|^2] \times \overline{\mathbb{E}_{T_{color}, X}}[|h_\theta(T_{color}(X)) - h_\theta(X)|^2] \qquad (2)$$

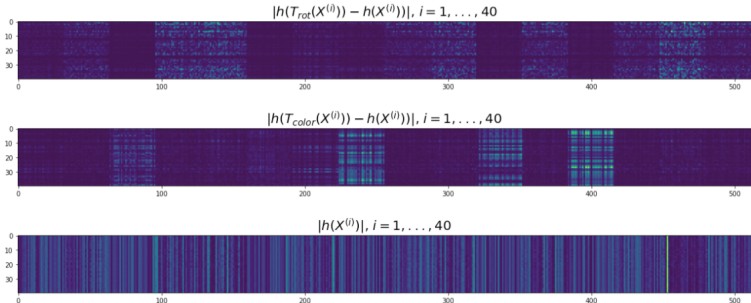

Figure 5: First row: stacked 512 dimensional vectors representing $|h_\theta(T_{rot}(x)) - h_\theta(x)|$ for random rotations $T_{rot}$. Second row: stacked 512 dimensional vectors representing $|h_\theta(T_{color}(x)) - h_\theta(x)|$ for random color rotations $T_{color}$. Third row: stacked 512 dimensional vectors representing $|h_\theta(x)|$. Dark (small value) regions represent the invariant spaces.

Table 1: Linear evaluation accuracy scores. For Digit, we conducted Linear Logistic Regression. For the prediction of color and angle, we conducted linear regression on the pair of images with random color and rotation to predict the color difference and angle difference.

| Features | Ours | SimCLR | Raw representation |
|---|---|---|---|
| Digit Accuracy (Logistic) | $0.5836 \pm 0.0013$ | $0.6091 \pm 0.0015$ | $0.5175 \pm 0.0027$ |
| Angle Prediction Error | $0.0102 \pm 0.0002$ | $0.0245 \pm 0.0023$ | $42.4783 \pm 31.6392$ |
| Color Prediction Error | $0.0037 \pm 0.0001$ | $0.0175 \pm 0.0001$ | $9.4576 \pm 4.5193$ |

Table 2: Linear classification result on the learned representations.

Where $\overline{A}$ designates the normalization of the vector $A$. By definition, this evaluates to 0 when the $T_{rot}$-sensitive space has empty intersection with the $T_{color}$-sensitive space. In our experiment, simCLR yielded the score of $0.0016 \pm 0.0001$, while our $L_1$ contrastive learning yielded the score of $0.00012 \pm 0.00001$, numerically validating the complementary decomposition we visually see in Figure 5.

Also, to verify whether our decomposed representation holds enough information to predict the important features of our dataset, namely the rotation angle, the color, and the digit shape, we conducted a linear regression for each one of these features based on the learned representation. Table 1 summarizes the result of our linear evaluation. For the angle and color, we encoded a pair of images $x_1, x_2$ with random color and random rotation, concatenated the encoded output $z_1, z_2$, and linearly predicted the color difference in RGB and the sine value of the angle difference. Because the theme of our study is the representation of the encoder output, we conducted the evaluation on the final output layer. As we see in Table 1, our model achieves competitive scores for all features, and predicts the color differences particularly well. The raw representation (flattened 32*32*3 vector of image pixels) performed poorly on the test evaluation. We shall also note that, although the latent represetation obtained from the simCLR-trained encoder (Figure 6) does not feature the decomposition like the one observed in Figure 5, it has the ability to predict the features like angle and color, possibly indicating the presence of a hidden invisible structure underneath the representation . To visually see if our representation retains enough information of the original images, we also trained decoder on the fixed encoder trained from our $L_1$ contrastive objective function. As we can see in Figure 7 our representation encodes strong information regarding the orientation, shape and color of the image while featuring the decomposability.

## 4 CONCLUSION

In this work, we have investigated the representation learned by the contrastive learning with $L_1$ type distance. We have shown that, without any auxiliary regularization, the $L_1$ contrastive loss has the ability to decompose the dataspace in such a way that the invariant space of any augmentaion transformation $T$ can be represented as a set of coordinates. Our study suggests that, by considering different metrics, we may uncover more secrets behind the mechanism of contrastive learning and the representation learning in general.

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

## A    PROOF OF THE PROPOSITION 1

Let $\mathcal{T} = \{T\}$ be a set of augmentation transformations on $\mathcal{X}$, and

$$h : \mathcal{X} \to \mathbb{R}^n$$

be a homeomorphism, which introduces a coordinate of $\mathcal{X}$. Let $W_T$ denote the invariant subspace of $T \in \mathcal{T}$ through $h$, that is, the maximal subspace of $\mathbb{R}^n$ such that $h \circ T \circ h^{-1}(v) = v$ holds for any $v$ in the space. For the augmentations $\mathcal{T}$, we can consider the $\pi$-system $\mathcal{S}$ generated by the $\{W_T\}_{T \in \mathcal{T}}$, that is, $\mathcal{S}$ is the family of subspaces which are generated by intersections any finite subsets of $\{W_T\}_{T \in \mathcal{T}}$. Let $\mathcal{V} = \{V_j\}_{j=1}^m$ be a family of subspaces in $R^n$. If $\oplus_{j=1}^m V_j = \mathbb{R}^n$ holds and any $W \in \mathcal{S}$ is expressed as the direct sum of finite elements of $\mathcal{V}$, we call $\mathcal{V}$ *symmetry adapted*. If $\mathcal{T}$ is a finite set, the symmetry adapted subspaces is given by the minimal intersections of $\{W_T\}_{T \in \mathcal{T}}$, where minimal should be interpreted in terms of inclusion of sets. Thus, the symmetry adapted family is uniquely determined.

**Proposition** (Formal Version). *Under the above notations, let $\{V_j\}_{j=1}^m$ be the symmetry adapted subspaces for $\mathcal{T}$, and assume that $\mathcal{T}$ acts transitively in the strong sense, that is, for any $k$, $v_k \in V_k$, and $v_{-k}, v'_{-k} \in V_{-k} := \bigoplus_{j \neq k} V_j$, there exists some $V_k$-fixing $T$ such that $T \circ h^{-1}(v_k, v_{-k}) = h^{-1}(v_k, v'_{-k})$. Then, $\{V_j\}$ are block identifiable, that is, if there is another $\tilde{h}$ with $\{\tilde{V}_j\}$ that satisfy the assumptions, then there exists an invertible map between $V_j$ and $\tilde{V}_j$ for each $j$ after appropriate reordering of $\{\tilde{V}_j\}$.*

*Proof.* Let $v$ denote an element of $R^n = \bigoplus V_i$, and let $v_{-i} \in \bigoplus_{j \neq i} V_j$ denote a vector with the $i$-th component removed from $v$. Note that $\phi = \tilde{h} \circ h^{-1} : \bigoplus_i V_i \to \bigoplus_i \tilde{V}_i$ is an invertible map. By the last remark before Proposition, there is a one-to-one correspondence between $\{V_j\}$ and $\{\tilde{V}_j\}$, so we assume w.l.o.g. that for each $k$, $V_k$ and $\tilde{V}_k$ have the same set of $T$s to which they are invariant. Now, write

$$\phi(v) = [\phi_k(v), \phi_{-k}(v)]$$

where $\phi_k$ is the $\tilde{V}_k$ component of $\phi$, or $\tilde{\pi}_k \circ \phi$ with the projection operator $\tilde{\pi}_k : \bigoplus_{j=1}^{m} \tilde{V}_j \to \tilde{V}_k$. We first show that $\phi_k(v)$ depends only on $v_k = \pi_k(v)$, that is,

$$\phi_k(v_k, v'_{-k}) = \phi_k(v_k, v_{-k})$$

for all $v_k \in V_k$ and $v_{-k}, v'_{-k} \in V_{-k}$. By the assumption, there exists a $V_k$-fixing $T$ such that $T \circ h^{-1}(v_k, v_{-k}) = h^{-1}(v_k, v'_{-k})$ for all $v_k, v_{-k}$ and $v'_{-k}$. Because $T$ is $V_k$-fixing, it is also $\tilde{V}_k$-fixing. Thus,

$$\tilde{\pi}_k \circ \tilde{h} \circ T \circ \tilde{h}^{-1} = \tilde{\pi}_k \circ T \circ \tilde{h}^{-1},$$

or we can say $\tilde{\pi}_k \circ \tilde{h} \circ T = \pi_k \circ \tilde{h}$. Putting all together, we have

$$\begin{aligned} \tilde{\pi}_k \circ \phi(v_k, v_{-k}) &= \tilde{\pi}_k \circ \tilde{h} \circ h^{-1}(v_k, v_{-k}) \\ &= \tilde{\pi}_k \circ \tilde{h} \circ T \circ h^{-1}(v_k, v_{-k}) \\ &= \tilde{\pi}_k \circ \tilde{h} \circ h^{-1}(v_k, v'_{-k}) \\ &= \tilde{\pi}_k \circ \phi(v_k, v'_{-k}), \end{aligned}$$

which shows that the map $\phi_k$ depends only on $v_k$. Write $\phi_k(v_k) := \phi_k(v)$, then

$$\phi(v) = \bigoplus_{k=1}^{m} \phi_k(v_k).$$

Since $\phi$ is invertible, the above relation guarantees that each $\phi_k$ is invertible, and the identifiability follows.

$\square$

## B  ADDITIOANL FIGURES

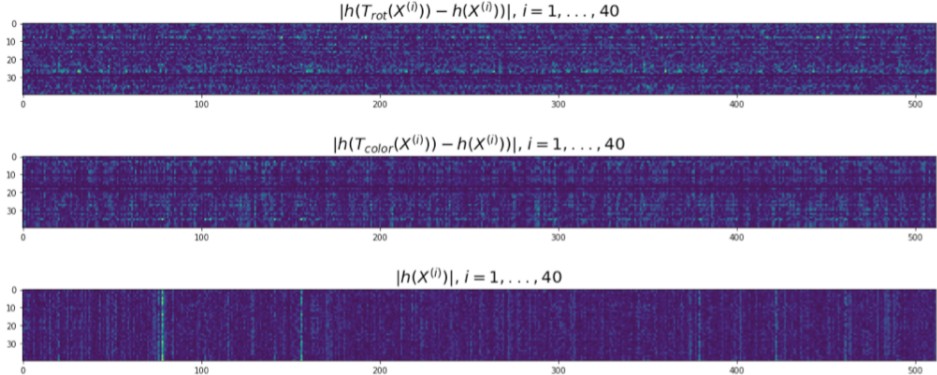

Figure 6: Visualization of SimCLR learned representation. First row: stacked 512 dimensional vectors representing $|h_\theta(T_{rot}(X)) - h_\theta(X)|$ for random rotations $T_{rot}$. Second row: stacked 512 dimensional vectors representing $|h_\theta(T_{color}(X)) - h_\theta(X)|$ for random color rotations $T_{color}$. Third row: Second row: stacked 512 dimensional vectors representing $|h_\theta(X)|$. No apparent structure is visible in simCLR representation.

## C  THE ARCHITECTURE USED IN STYLED MNIST DATASET.

For the experiment in Sec.3.2, we adopted ResNet-based encoder and decoder architecture (He et al., 2016). We use ReLU function (Nair & Hinton, 2010; Glorot et al., 2011; Maas et al., 2013) for each activation function and the group normalization (Wu & He, 2018) for the normalization layer. The details of the architecture is found in Figures 8 and 9.

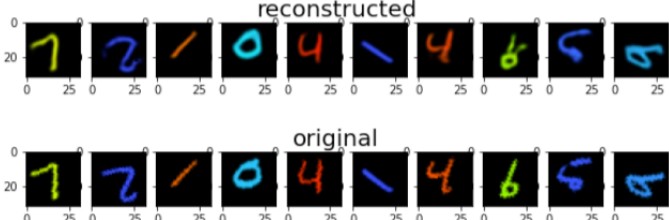

Figure 7: We trained an decoder for the encoder trained with our $L_1$ contrastive loss. We see that our encoder with decomposed feature dimensions retains much information about the digit shape, orientation and color.

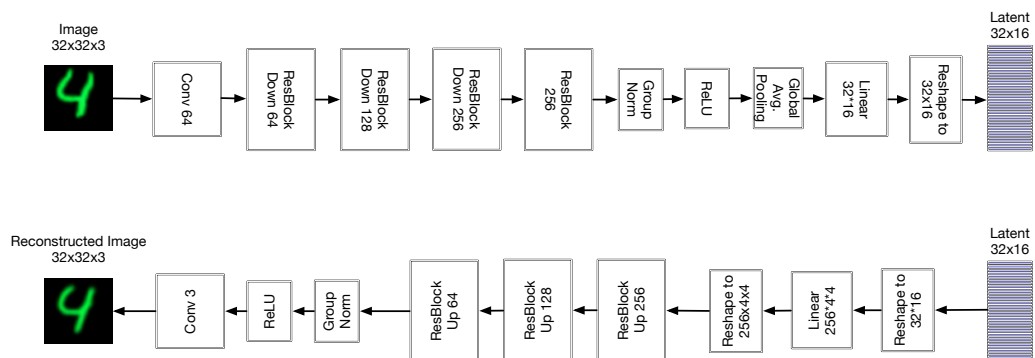

Figure 8: The encoder (top) and decoder (bottom) architecture we used in our experiments. The detail of the ResBlock architecture is described in Fig.9. The kernel size of all of the convolution layers is set to 3x3 except the layers replacing the skip connection in the ResBlocks, which is set to 1x1. Each number following each layer name indicates the output dimension for the linear layer and output channels for the convolution layer.

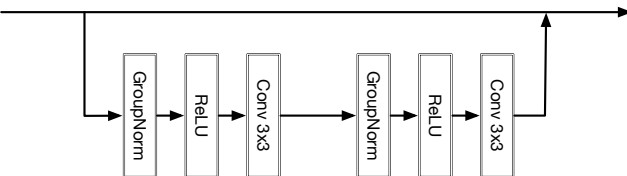

Figure 9: The detail of the ResBlock architecture. For the upsampling and downsampling, we followed the same procedure in Miyato & Koyama (2018). For downsampling, we replaced the identical mapping with 1x1 convolution followed by downsampling layer (mean avegrage pooling). For upsampling, we also replaced the identical mapping with nearest-neighbor upsampling followed by 1x1 convolution. The number of groups for the group normalization layer was set to 32. Weight standarization (Qiao et al., 2019) is applied to each 3x3 convolution layer.

