# OpenReview forum: "Decomposition into invariant spaces with $L_1$-type contrastive learning"
_ICLR.cc/2022/Workshop/OSC — Submitted to ICLR2022 OSC _

### Official Review · Reviewer_fQsB · 2022-03-12
**Good workshop paper on contrastive learning with L1 losses**

**Rating:** 2
**Confidence:** 2

**Review:**

This paper suggests that by changing cosine-similarity SimCLR-like contrastive learning losses with L1-losses, a decomposition of the learned latent spaces along different data augmentations can be achieved. First, the paper provides some theoretical guarantees for such block-identifiability under some rather special assumptions. Second, the effect of using L1-contrastive losses is being empirically investigated under a simple linear toy experiment as well as an experiment involving the stylized MNIST dataset.

Pros:
- The paper is well written and easy to follow.
- The presented experiments, analysis and evaluations are reasonable and solid, supporting the main claim of the paper.
- Discussion of theoretical background and related recent works.
- Interesting empirical results on the MNIST experiment, especially the clear decompositions of the learned latent spaces.

Weaknesses/Suggestions
- Regarding the MNIST experiments, I would have appreciated to see some further justification in choosing a latent feature map and the generalized group-lasso objective. Why did you decide for that and not for a standard 128-dimensional latent space (which seems sufficient for MNIST) and a standard L1 loss as in the original SimCLR implementation?
- While I agree that it is interesting to see the improved downstream prediction of the augmentation features, the SimCLR model is only slightly worse while being better at the digit prediction. I would be curious to see how this changes when using a 2-layer or 3-layer MLP as prediction model that might be better at inferring this information from the SimCLR representation.
- I think it would be great to see some experiments, which show some practical benefits of the obtained representations with such decomposed structure,  i.e. the representations’ robustness and OoD generalization abilities compared to the performance of the SimCLR model.


Minor:
The table seems to have two captions: “Table 1” and “Table 2”

---

### Official Review · Reviewer_57QJ · 2022-03-16
**Decomposition into invariant spaces with L1-type contrastive learning**

**Rating:** 1
**Confidence:** 2

**Review:**

Overall: While the work is interesting, and there could certainly be a novel contribution here, I cannot in good faith recommend acceptance as due to the lack of clarity, I am not confident in my technical correctness check. I recommend the authors use the appendix to spell out all details regarding both the theory and the experiments, s.t. the proof can be easily verified, and the experiments are truly reproducible.

Specific comments:
-Proposition 1 can be considered to be quite vague. As a start, it would be helpful to clarify what is implied by the symbols, i.e. $\tilde{h}$ vs $h$, if anything.
-Could the authors clarify this statement, "We note that this statement assumes that the set of augmentations defining the frequencies is rich enough that there is no frequency that is not fixed by any augmentation." Given the reader has just been introduced to the concepts in the paper, it would be helpful if this statement could be extended to provide any clarifying details.
-Furthermore, while it is implied the identifiability criterion is the same as (von Kugelgen et al., 2021), given the distinction between this work and (von Kugelgen et al., 2021), the identifiability criterion should be written explicitly, and the proof should conclude with a result which implies said statement.
-Authors should note the similarities (and differences) between eq. (1) and Definition 1 in (Zimmermann et al. 2021). Specifically, note that Theorem 6 in (Zimmermann et al. 2021) proved identifiability up to permutation for an objective which uses $L_1$ for the similarity measure, albeit with a set of assumptions. Furthermore, see "Laplace" under model assumptions in Tables 2+3 for empirical validation.
-Authors should include the details regarding the data generating process in the appendix with equations, the description given in the main text is somewhat understandable, but again is a bit vague. For example, "However, hidden in the shades of unknown encoder map..., such an underlying structure may be hard to obtain just with the pairs...", highlighted vague verbage includes "hidden in the shades" and "hard".
-The precise mathematical details regarding the augmentation are pivotal for distinguishing between this work and previous work, it is not sufficient to have a clarifying figure, the precise formulas used should be included in the appendix.
-The authors imply they use the same identifiability criterion as (von Kugelgen et al., 2021), but in experiments, state they aim to achieve identifiability up to a permutation of the coordinates and scaling. Given they are speaking about the permutation of the subspaces, the relationship between this and the criterion in (von Kugelgen et al., 2021) can be understood, but, there remains ambiguity, as on face value, this verbage resembles the traditional "strong identifiability" criterion used in nonlinear ICA works, which includes (Zimmermann et al. 2021), which is especially notable given the aforementioned similarities between eq. (1) and Definition 1. Once again, ambiguity must be clarified with precise statements.
-Figure 3 appearing to indicate the desired correspondence, with statements like "We see that the size of the blocks in the second row are almost the same as the blocks in the blocks in the first row", is insufficient as a verification. A quantitative metric which corresponds to the precise identifiability criterion should be explicitly stated, and the scores should be computed.
-Section 3.2 is extremely unclear, what is $d_a$ and $d_s$? Hard to give detailed comments for this section, it is very unclear what exactly is the experiment here.
-Appendix A: Would be beneficial if the authors clearly stated their assumptions in a list format, s.t. assumptions can be referenced like "Assumption 1/A". Would help the readability of the proof. Specifically, "By the assumption, there exists a..." By which assumption?
-Check citations, for example, the wrong venue is listed for Zimmermann et al. 2021.
-Minor grammatical checks, i.e. "an unique" -> "a unique", "by intersections any finite subsets"

---

### Decision · Program_Chairs · 2022-03-24

**Decision:**

Reject

**Comment:**

Unfortunately, the reviewers found the paper to be not ready for presentation at the workshop. I recommend the authors to take a closer look at the reviews, in particular, reviewer 57QJ raises major concerns in the clarity. The paper has the potential to be improved by taking them into account.